# MINDFORMER: SEMANTIC ALIGNMENT OF MULTI-SUBJECT FMRI FOR BRAIN DECODING

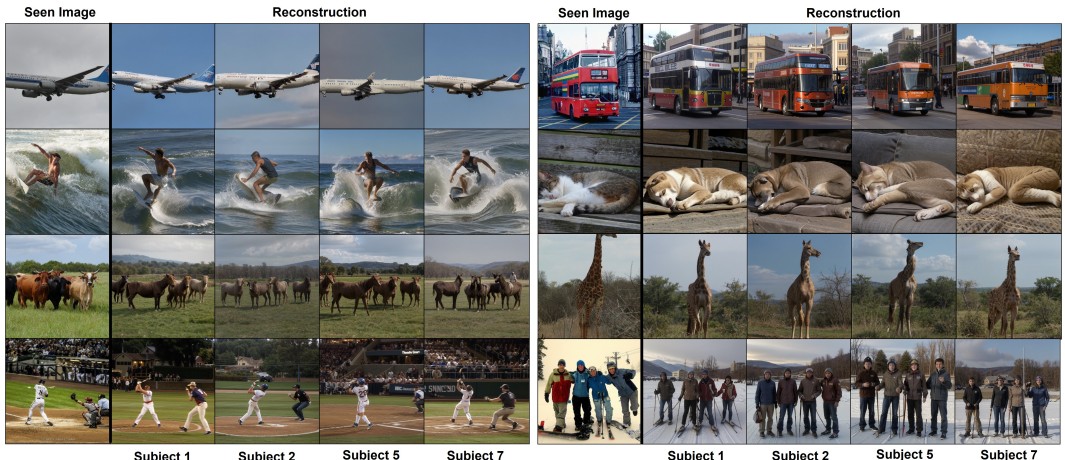

Figure 1: **Multi-subject brain decoding results by MindFormer.** MindFormer can reconstruct semantically aligned images across subjects. Additional reconstruction samples can be found in Figure 4 and Appendix A.2.

## ABSTRACT

Research efforts for visual decoding from fMRI signals have attracted considerable attention in research community. Still multi-subject fMRI decoding with one model has been considered intractable due to the drastic variations in fMRI signals between subjects and even within the same subject across different trials. To address current limitations in multi-subject brain decoding, here we introduce a novel semantic alignment method of multi-subject fMRI signals using so-called *MindFormer*. This model is specifically designed to generate fMRI-conditioned feature vectors that can be used for conditioning Stable Diffusion model for fMRI-to-image generation or large language model (LLM) for fMRI-to-text generation. More specifically, MindFormer incorporates two key innovations: 1) a subject specific token that effectively capture individual differences in fMRI signals while synergistically combines multi subject fMRI data for training, and 2) a novel feature embedding and training scheme based on the IP-Adapter to extract semantically meaningful features from fMRI signals. Our experimental results demonstrate that MindFormer generates semantically consistent images and text across different subjects. Since our MindFormer maintains semantic fidelity by fully utilizing the training data across different subjects by significantly surpassing existing models in multi-subject brain decoding, this may help deepening our understanding of neural processing variations among individuals.

## 1 INTRODUCTION

Brain decoding is a field dedicated to interpreting neural activity patterns to understand cognitive and sensory processes (Chen et al., 2014; Défossez et al., 2023; Du et al., 2022; Prince et al., 2022; Rao & Ballard, 1999; Schoenmakers et al., 2013). By utilizing neuroimaging techniques such as

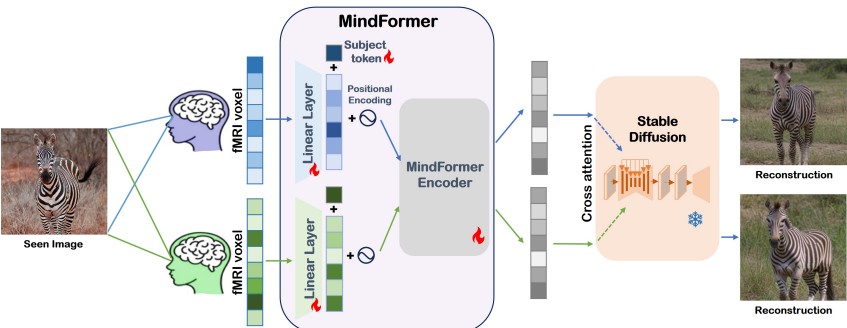

Figure 2: **MindFormer architecture**. The fMRI voxels obtained from observing the stimulus image are processed through the MindFormer to extract image features. These features are then utilized in conjunction with the Stable Diffusion model and a decoder to reconstruct the previously viewed image. MindFormer is trained to counter the subject specific bias through learnable subject token.

functional magnetic resonance imaging (fMRI), researchers measure brain activity in response to cognitive and sensory stimuli. A rapidly advancing area within this field is visual brain decoding, which aims to interpret neural signals to reconstruct visual experiences. The advent of deep learning has significantly propelled this field forward (Beliy et al., 2019; Gaziv et al., 2022; Gu et al., 2022; Horikawa & Kamitani, 2017; Shen et al., 2019; VanRullen & Reddy, 2019). One prevalent approach in visual decoding involves mapping neural activity to the latent spaces of generative models, such as generative adversarial networks (GANs) (Lin et al., 2022; Mozafari et al., 2020; Ozcelik et al., 2022; Seeliger et al., 2018). Recent advancements, fueled by new large-scale fMRI datasets (Allen et al., 2022), have seen the emergence of diffusion models, which enhance reconstruction accuracy (Chen et al., 2023; Lu et al., 2023; Mai & Zhang, 2023; Ozcelik & VanRullen, 2023; Scotti et al., 2024; Takagi & Nishimoto, 2023a; Xia et al., 2024; Wang et al., 2024). The integration of diffusion models in brain decoding marks a significant leap forward, providing advanced tools to reconstruct and interpret complex neural representations. Nonetheless, challenges remain in achieving high-fidelity reconstructions, lightweight models, and integrated subject-specific brain decoding.

In this work, we introduce a transformer-based multi-subject semantic alignment algorithm called MindFormer, which demonstrates exceptional performance in multi-subject brain decoding, particularly when combined with diffusion models or LLMs. MindFormer is specifically designed to generate semantically meaningful feature embeddings across multiple subjects to Stable Diffusion for image generation and LLMs for text generation. More specifically, to effectively integrate training data from multiple subjects while accounting for individual differences, we introduce a learnable subject token as inputs in the Transformer prompt. These components allow MindFormer to obtain semantically meaningful embedding even from limited datasets by leveraging collective information across subjects, improving its practical applicability in scenarios where data availability is restricted. Furthermore, we employ the IP-adapter, as described in Ye et al. (2023), to generate 16x768-dimensional feature embeddings from fMRI signals, which serve as conditioning inputs for the Stable Diffusion or LLMs. Unlike previous approaches that utilized CLIP embeddings, our use of the IP-adapter yields smaller, more efficient semantic embeddings, which reduces both computational costs and the risk of overfitting, and significantly enhances decoding accuracy and reliability. Experimental results demonstrate that our method maintains strong performance, even with limited data, by effectively utilizing shared information across subjects to maximize accuracy and reliability.

Our contributions can be summarized as follows:

- We developed a semantic alignment method of multi-subject fMRI data using MindFormer to effectively integrate training data from multiple subjects while accounting for individual differences by using learnable subject token as inputs in the Transformer prompt. These components allow MindFormer to generate semantically meaningful condition embedding modules for the Stable Diffusion model, specifically tailored for multi-subject brain decoding from fMRI signals.

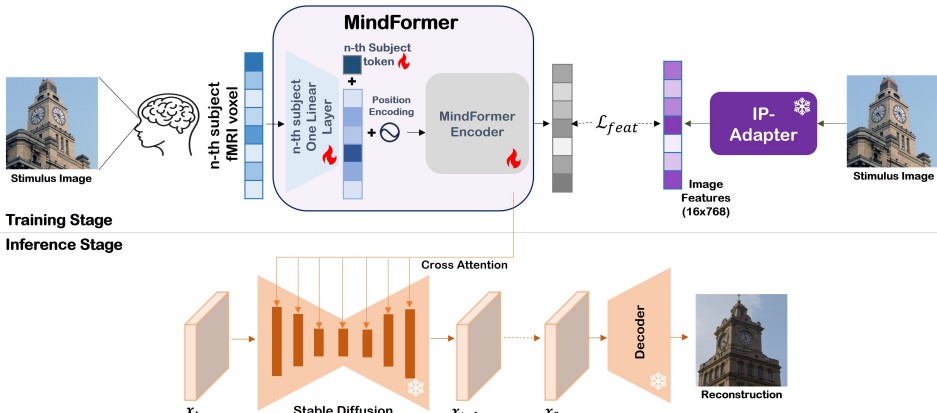

Figure 3: **Training stage**: The fMRI signals from each subject are passed through a subject-specific linear layer. Subsequently, each signal is prepended with a learnable subject token and passed through the same MindFormer Encoder. The network is then trained to match the image feature embeddings obtained from passing the images through the IP-Adapter. **Inference Stage**: These obtained embeddings are integrated into the stable diffusion process as conditions. The diffusion model utilizes these embeddings to iteratively denoise and reconstruct the image.

- Unlike previous methods that map brain signals into large CLIP image and text embeddings with dimensions of 257×768 and 77×768, respectively, MindFormer utilizes the IP-adapter to transform brain voxels into a more compact 16×768-dimensional space. Along with a lightweight, subject-specific linear layer and a learnable subject token, the use of the IP-adapter significantly reduces the overall model size. These optimizations make MindFormer substantially more efficient than existing models.

- To validate our method, we conducted experiments using the publicly available NSD dataset (Allen et al. (2022)). Experimental results confirm that the proposed method achieves excellent performance in multi-subject brain decoding. Also, our method effectively reconstructs high-quality images from limited datasets by leveraging shared information across subjects, maintaining strong performance even with constrained data availability.

- We demonstrate the universality of MindFormer embedding by showing that its embedding can be used for LLM as inputs for accurate fMRI-to-text generation.

## 2 RELATED WORKS

### 2.1 fMRI-TO-IMAGE RECONSTRUCTION MODELS

fMRI-to-image reconstruction models are advanced approaches designed to translate brain activity, captured through functional magnetic resonance imaging (fMRI), into visual images. These models learn complex mappings between neural signals and visual representations, allowing for the generation of images that closely resemble the original stimuli perceived by subjects. With the advent of deep learning, these models have increasingly leveraged deep learning frameworks to interpret and reconstruct visual experiences based on neural activity patterns. Recent advancements have integrated generative models, such as Generative Adversarial Networks (GANs) (Lin et al., 2022; Mozafari et al., 2020; Ozcelik et al., 2022; Seeliger et al., 2018) and Variational Autoencoders (VAEs) (Han et al., 2019), with fMRI data to improve the accuracy and quality of reconstructed images. For instance, Seeliger et al. (2018) explored the use of GANs for fMRI-to-image synthesis, while Han et al. (2019) demonstrated the effectiveness of VAEs in reconstructing visual stimuli from brain activity. Advances in deep learning have enabled the reconstruction of not only natural scenes but also human faces (Dado et al., 2022; VanRullen & Reddy, 2019) and video stimuli Wang et al. (2022). Additionally, techniques like contrastive learning (Chen et al., 2020; Radford et al., 2021) have been employed to better align neural embeddings with visual embeddings. This alignment significantly enhances the fidelity of the reconstructed images, ensuring that the generated visuals accurately reflect the subjects' visual experiences.

## 2.2 IMAGE GENERATION DIFFUSION MODEL

Another significant innovation in brain decoding is the use of diffusion models. Diffusion models have gained popularity in generative modeling due to their ability to transform noise vectors into output images through a reverse diffusion process (Ho et al., 2020; Nichol & Dhariwal, 2021; Song et al., 2020a). Recent studies (Dhariwal & Nichol, 2021; Song et al., 2020b) have demonstrated that diffusion models achieve superior image generation quality compared to GANs (Brock et al., 2018; Zhang et al., 2019), further establishing their importance in this field. These models have been particularly impactful in brain decoding, offering enhanced flexibility and precision in capturing the subtle nuances of visual experiences encoded in brain activity (Chen et al., 2023; Lu et al., 2023; Mai & Zhang, 2023; Ozcelik & VanRullen, 2023; Scotti et al., 2024; Takagi & Nishimoto, 2023a; Xia et al., 2024). In brain decoding studies that utilize diffusion models, researchers often employ the Stable Diffusion (Rombach et al., 2022) or Versatile Diffusion (Xu et al., 2023) models. Stable Diffusion focuses on generating high-quality images by refining noise into coherent visuals, while Versatile Diffusion enables substantial image variation, facilitating tasks like style transfer. However, this variability in Versatile Diffusion can introduce challenges in brain decoding, as the reconstructed images may display inconsistencies and artifacts. These discrepancies can complicate the accurate interpretation of neural activity, potentially reducing the fidelity of the decoding outcomes.

Recent advancements in controllable image generation, such as ControlNet (Zhang et al., 2023) and T2I-adapter (Mou et al., 2024), have shown that additional networks can guide image generation by plugging into existing text-to-image diffusion models. However, these methods often fall short in faithfully reproducing the reference image, primarily due to limitations in the cross-attention modules that inadequately merge image and text features. IP-Adapter (Ye et al., 2023) addresses this issue by effectively integrating image features, enabling more accurate and detailed image generation based on reference inputs. The central concept of the IP-Adapter revolves around its decoupled cross-attention mechanism. Instead of employing a single cross-attention layer to handle both text and image features simultaneously, the IP-Adapter introduces a dedicated cross-attention layer specifically for image features. This separation enables the model to focus on learning more detailed and image-specific features, enhancing its ability to capture the unique characteristics of visual data.

## 3 MINDFORMER

### 3.1 MODEL ARCHITECTURE

As shown in Fig. 2, the fMRI signals obtained during the $n$-th subject's viewing of an image are initially passed through a subject-specific linear layer. Following this, each signal is prepended with a unique learnable subject token and then processed through the MindFormer encoder. The embeddings produced from this process are subsequently trained to match the image feature embeddings obtained when the images are passed through the IP-Adapter. It is important to note that all subjects' signals are processed through the same instance of the MindFormer encoder, ensuring a unified encoding process. For the case of image generation, the trained embedding, from the brain signal, are integrated into the Stable Diffusion process. The diffusion model utilizes these embeddings to iteratively denoise and reconstruct the image to generate semantically aligned images.

Specifically, as shown in Fig. 3, MindFormer comprises of the subject specific linear layer and a single transformer encoder that incorporates unique learnable subject token. The architecture of MindFormer encoder follows the Vision Transformer (ViT) (Dosovitskiy et al., 2020) encoder. Note that fMRI signals vary in size across subjects, primarily due to inherent differences in brain size and structure. To address the differing input voxel sizes, MindFormer maps each subject's voxels $v^s$ into a uniform dimension of 16×768 through individual linear layers $\mathcal{E}_s$ for each subject $s$. Then, in the position of BERT (Devlin et al., 2018) and ViT (Dosovitskiy et al., 2020)'s [Class] token, we prepend a learnable embedding token $x_{subj}$ to the output $x^s = \mathcal{E}_s(v^s)$ of linear mapping. Addition use of learnable subject token is intended to decouple the individual bias of fMRI signal differences from the common representation across subjects, thereby allowing accurate interpretation of the neural data corresponding to multiple subject as well as each individual subject. Then, the position embeddings $P$ are incorporated into the prepared embeddings to preserve positional information. The following steps proceed similarly to the transformer encoder in ViT. The Transformer encoder is composed of alternating layers of multi-headed self-attention (MSA) and MLP blocks. Layer normalization (LN)

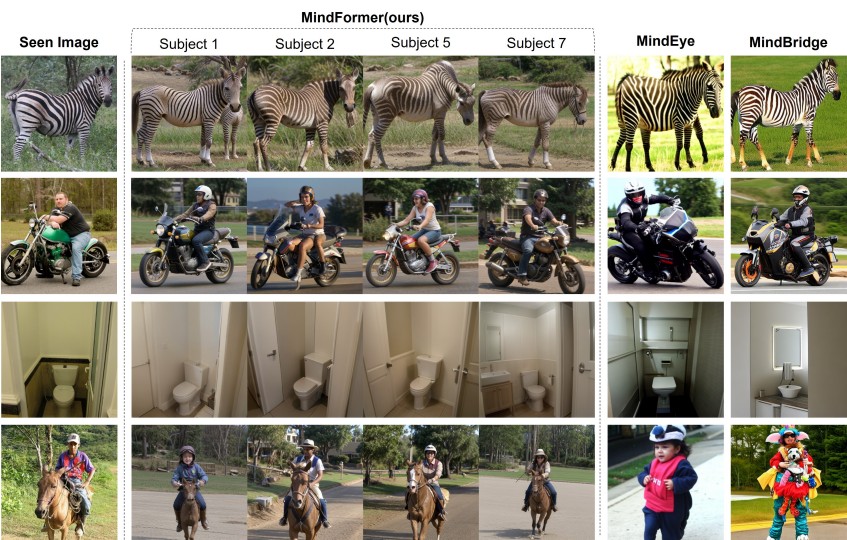

Figure 4: **Visual comparison of our proposed MindFormer with other methods.** Our resulting images are semantically closest to the seen images.

is applied before each block, with residual connections following each block. The MLP consists of two layers with a GELU activation function.

## 3.2 TRAINING OBJECTIVES

Formally, we represent the 1D fMRI voxels from subject $s$ as $\boldsymbol{v}^s \in \mathbb{R}^{F_s}$, where $F_s$ denotes the subject specific size of the fMRI voxels. The corresponding image stimulus $I$ can be extracted as image feature embeddings $\mathbb{E}_I = [\boldsymbol{e}_1, \cdots, \boldsymbol{e}_N] \in \mathbb{R}^{d \times N}$ using a pretrained IP-Adapter with $N = 16$ and $d = 768$. Additionally, MindFormer maps the brain voxels $\boldsymbol{v}^s$ into a 16×768-dimensional vector $\boldsymbol{Z} = [\boldsymbol{z}_1, \cdots, \boldsymbol{z}_N]$, matching the dimension of the image feature embeddings $\mathbb{E}_I$ from the IP-Adapter (Ye et al., 2023). Then, MindFormer is trained with feature domain $l_1$-loss and contrastive learning loss between fMRI and Images:

$$\mathcal{L}_{feat} = \mathcal{L}_1 + \alpha \cdot \mathcal{L}_{contrastive} \tag{1}$$

where $\alpha > 0$ is a weight parameter. Specifically, the image feature-domain $l_1$ loss measures how MindFormer can predict the image feature $\mathbb{E}_I$ from IP-Adapter:

$$\mathcal{L}_1(\boldsymbol{Z}, \mathbb{E}_I) = \frac{1}{N} \sum_{i=1}^{N} \|\boldsymbol{z}_i - \boldsymbol{e}_i\|_1 \tag{2}$$

The contrast loss imposes the structural similarity between the MindFormer's output and that of IP-Adapter by increasing the similarity between the feature at the same location while decreasing the similarity at different loations:

$$\mathcal{L}_{contrastive}(\boldsymbol{Z}, \mathbb{E}_I) = \frac{1}{N} \sum_{i=1}^{N} \left( \log \frac{e^{\boldsymbol{z}_i \cdot \boldsymbol{e}_i}}{\sum_{j=1}^{N} e^{\boldsymbol{z}_i \cdot \boldsymbol{e}_j}} \right) \tag{3}$$

## 4 EXPERIMENTS RESULTS

**Experiment Settings.** The proposed model is implemented in PyTorch. The single subject model is trained on one NVIDIA RTX 3090 GPU with a 24GB memory, and the multi subject model is trained on one NVIDIA RTX V100 with a 32GB memorys. Across all experiments, the batch size is set to 4 per GPU and the epoch size is 50. The learning rate was set as $3 \times 10^{-4}$, and the

| Method | # Models | Low-level | | | | High-level | | | |
|---|---|---|---|---|---|---|---|---|---|
| | | PixCorr ↑ | SSIM ↑ | Alex(2) ↑ | Alex(5) ↑ | Incep ↑ | CLIP ↑ | EffNet-B ↓ | SwAV ↓ |
| Takagi et al. | 4 | – | – | 83.0% | 83.0% | 76.0% | 77.0% | – | – |
| Brain-Diffuser | 4 | .254 | **.356** | 94.2% | 96.2% | 87.2% | 91.5% | .775 | .423 |
| MindEye | 4 | **.309** | .323 | **94.7%** | **97.8%** | 93.8% | 94.1% | **.645** | .367 |
| MindBridge (Single-) | 4 | .148 | .259 | 86.9% | 95.3% | 92.2% | 94.3% | .713 | .413 |
| Ours (Single-) | 4 | .241 | .352 | 93.5% | 97.5% | 93.5% | 93.6% | .659 | .356 |
| MindBridge | 1 | .151 | .263 | 87.7% | 95.5% | 92.4% | **94.7%** | .712 | .418 |
| Ours (Multi-) | 1 | .243 | .345 | 93.5% | 97.6% | **94.4%** | 94.4% | .648 | **.350** |

Table 1: **Quantitative comparison of MindFormer's decoding performance against other models**. The metrics presented are averaged across the data from 4 subjects. Unlike other methods, which generally require a separate model for each subject, our approach and MindBridge consolidate the process into one model. Among them, our approach achieved superior results in all metrics except for the CLIP score. **Bold**: best, underline: second best.

moment parameters of the AdamW optimization algorithm (Loshchilov & Hutter, 2017) were set as $\beta_1 = 0.9, \beta_2 = 0.999$.

**Dataset.** To better understand the task at hand, we illustrate the data used in our study. For all experiments, we used the widely-adopted Natural Scenes Dataset (NSD) (Allen et al., 2022), a public fMRI dataset containing high-resolution 7-Tesla fMRI scans of brain responses from eight healthy adult subjects viewing natural scenes from the MS-COCO dataset (Lin et al., 2014). Following common practices (Mai & Zhang, 2023; Ozcelik & VanRullen, 2023; Scotti et al., 2024; Takagi & Nishimoto, 2023a; Wang et al., 2024), our research primarily uses data from four subjects (subj01, 02, 05, 07) who completed all scan sessions. Specifically, only a subset of data—982 images—was commonly viewed by all four subjects and used as the test set. The remaining data, comprising 8,859 distinct images viewed by each subject, were used as the training set, resulting in 24,980 training samples without averaging across repetitions, similar to the method used by previous research. We utilize the dataset, preprocessed by Scotti et al. (2024), which consists of flattened fMRI voxels within the brain volume space corresponding to the "nsdgeneral" brain region. This region, defined by the authors of Allen et al. (2022), includes the subset of voxels that are most responsive to visual stimuli.

## 4.1 EXPERIMENTAL RESULTS

To quantitatively compare with other methods, we utilize eight image quality evaluation metrics as outlined in Ozcelik & VanRullen (2023). For assessing low-level properties, we use PixCorr, SSIM (Wang et al. (2004)), AlexNet(2), and AlexNet(5) (Krizhevsky et al. (2012)). For evaluating higher-level properties, the metrics of Inception (Szegedy et al. (2016)), CLIP (Radford et al. (2021)), EffNet-B (Tan & Le (2019)), and SwAV (Caron et al. (2020)) are employed. We compared our model with Takagi & Nishimoto (2023a), Brain-Diffuser (Ozcelik & VanRullen (2023)), MindEye (Scotti et al. (2024)), and MindBridge (Wang et al. (2024)).

Figure 1 demonstrates MindFormer's strong performance across all four subjects, consistently producing accurate and reliable results. From the images, the effectiveness of the model is evidenced by its ability to generalize well and maintain high accuracy in decoding brain activity into visual images for each subject. The reconstructed images, shown in Figure 4, clearly illustrate the superior performance of our approach compared to existing methods. The results from the proposed method highlights the accuracy and fidelity of the visual outputs generated by our model, aligning closely with the original stimuli. In particular, our model's results demonstrate a high degree of semantic similarity to the stimulus images. This is evident in the ability of our model to accurately capture and reproduce the high-level features present in the original stimuli, resulting in reconstructed images that closely resemble the meaning and content of the stimulus images. This high semantic fidelity highlights the effectiveness of our approach in maintaining the integrity of the visual information during the decoding process.

Also, the quantitative metrics presented in Table 1 further support these findings, indicating significant improvements in high-level indicators such as Inception, CLIP, EffNet-B and SwAV. In brain decoding, low-level metrics evaluate the pixel-wise and structural similarity between original and reconstructed images. On the other hand, the high-level metrics assess the semantic similarity and how well the

Figure 5: Reconstructed image from human brain activity on the presence of learnable subject tokens (ST) in MindFormer. The model incorporating subject tokens demonstrates higher correlation in semantic meaning with the seen image.

| Subject Token (ST) | Low-level | | | | High-level | | | |
|---|---|---|---|---|---|---|---|---|
| | PixCorr ↑ | SSIM ↑ | Alex(2) ↑ | Alex(5) ↑ | Incep ↑ | CLIP ↑ | EffNet-B ↓ | SwAV ↓ |
| without ST | .233 | **.349** | 92.8% | 97.1% | 93.4% | 93.4% | .662 | .359 |
| with ST | **.243** | .345 | **93.5%** | **97.6%** | **94.4%** | **94.4%** | **.648** | **.351** |

Table 2: **Quantitative comparison of results on the presence or absence of learnable subject tokens (ST)**. The inclusion of subject tokens significantly improved the model's accuracy, as evidenced by the higher scores across various evaluation metrics. This demonstrates that learnable subject tokens play a crucial role in semantic alignment of multi-subject fMRI signal and the reliability of the brain decoding process. **Bold**: best.

reconstructed images capture the meaning and content of the originals. High-level metrics being high indicates that a model effectively captures and reconstructs the complex, abstract features of the stimulus, leading to more meaningful and contextually accurate representations. Therefore, for the purpose of semantically aligned brain decoding from multi-subject data, high-level metric is more important. Thus, the results in Table 1 confirm the effectiveness of our model in semantically aligned decoding from neural data.

Additionally, not only does our model achieve high scores on high-level metrics, but it also consistently outperforms other models trained on data from multiple subjects, such as MindBridge, in terms of low-level metrics as demonstrated in Table 1. This table illustrates that our model attains superior scores across a range of metrics when compared to MindBridge, which is also designed to handle data from multiple subjects within a single model. These results suggest that our model is more effective in multi-subject fMRI signal alignment and decoding, thereby implying its potential for broader adoption and application in the field. By excelling in both high-level semantic representation and overall performance, our model demonstrates a significant advancement in the ability to decode and reconstruct brain activity from multiple individuals within a unified framework.

## 4.2 ABLATION STUDY

**Importance of subject tokens.** Using learnable subject tokens in MindFormer has several positive effects. First, it allows the model to accurately distinguish between inputs from different subjects, ensuring that individual-specific neural patterns are correctly interpreted and processed. This enhances the precision of brain decoding by aligning the model's understanding with the unique characteristics of each subject's brain activity. Additionally, the incorporation of learnable subject tokens improves the model's ability to generalize across multiple subjects, as it can adapt to variability in neural signals while maintaining high performance in decoding tasks. Overall, subject tokens contribute to more reliable and robust decoding outcomes, facilitating better insights into neural representations. Figure 5 and Table 2 provides evidence for these benefits by showing superior performance results when subject tokens are used. Also, by using subject tokens, this unified model approach offers significant advantages in terms of efficiency and scalability, allowing for comprehensive analysis and image reconstruction across different individuals without the need for multiple models.

**Exploiting multiple subject data set.** Given the limited training data, obtaining a sufficiently large fMRI dataset from new subjects is a challenging task. Therefore, it is crucial to achieve high-quality result images even with a small amount of data. To investigate this, we perform ablation study using single-subject and multi-subject experimental setups. In single subject scenario, MindFormer trains only the dataset of Subject 1. In multi subject scenario, the training process includes not only subject 1 but also subjects 2, 5, and 7. Data from each of these subjects is processed through the MindFormer

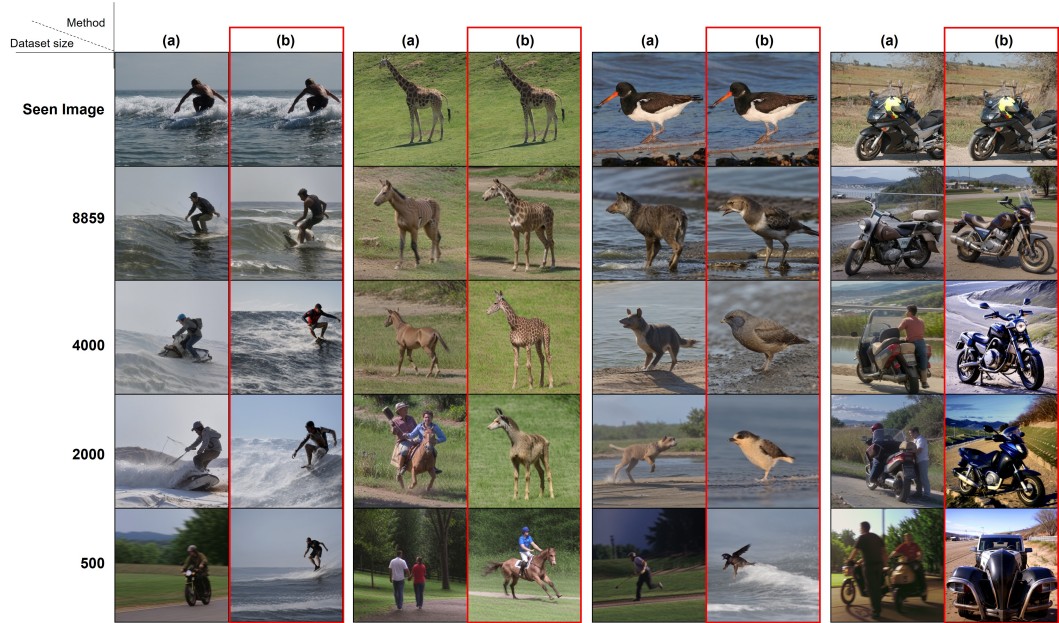

Figure 6: **Performance comparison on limited datasets**. With limited training data from a single subject, our proposed MindFormer can reconstruct natural images more accurately by leveraging knowledge from other subjects. The results shown are for Subject 1, trained on different dataset size. Two scenarios are compared: (a) single subject scenario, where the MindFormer is trained exclusively on subject 1's data, and (b) multi subject scenario, where the MindFormer is trained on data from subjects 1, 2, 5, and 7.

framework, allowing the model to learn from a varied set of neural patterns. This ablation study investigates how the model can generalize more effectively across different individuals, thereby enhancing its decoding accuracy and robustness.

Figure 6 and Table 3 presents the reconstructed images and metric results for subject 1 across different dataset sizes: the entire dataset, 4000 samples, 2000 samples, and 500 samples. Overall, the results indicate that the multi-subject approach consistently outperforms the single subject approach. Notably, the results obtained using only 2000 samples demonstrate that multi-subject training, which is our method, can effectively reconstruct high-quality images even with a limited amount of data. For example, even with as low as 500 samples, multi-subject approach still outperforms the single subject approach. Specifically, Figure 6 shows that as the dataset size decreases, the single subject approach struggles to preserve the semantic aspects of the stimulus image, whereas the multi subject approach maintains this semantic fidelity well. This highlights the robustness and efficiency of multi MindFormer in leveraging small datasets to achieve superior image reconstruction. By incorporating multiple subjects, MindFormer can leverage the collective information, leading to improved performance in brain decoding tasks.

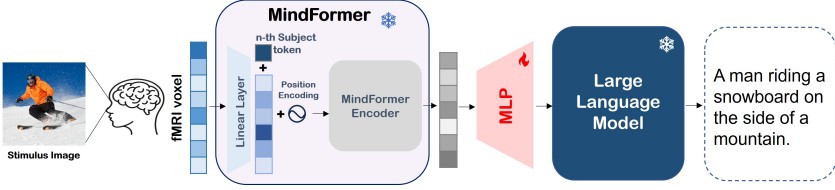

Figure 7: **fMRI-to-Text Model using a pretrained Mindformer and an LLM**. For the input of LLM, the output of the MindFormer is mapped to textual embedding using a two-layer MLP.

| Method | # Dataset | Low-level | | | | High-level | | | |
|---|---|---|---|---|---|---|---|---|---|
| | | PixCorr ↑ | SSIM ↑ | Alex(2) ↑ | Alex(5) ↑ | Incep ↑ | CLIP ↑ | EffNet-B ↓ | SwAV ↓ |
| (a) | all | .270 | **.356** | 95.2% | 98.0% | 93.4% | 93.6% | .660 | .354 |
| (b) | all | **.271** | .351 | **95.3%** | **98.2%** | **95.1%** | **94.7%** | **.641** | **.339** |
| (a) | 4000 | **.264** | **.369** | **93.5%** | 96.9% | 91.6% | 92.3% | .701 | **.374** |
| (b) | 4000 | .227 | .327 | 93.0% | **97.5%** | **93.6%** | **93.7%** | **.680** | .392 |
| (a) | 2000 | .221 | .344 | 90.6% | 95.6% | 88.6% | 89.1% | .757 | **.401** |
| (b) | 2000 | **.241** | **.352** | **92.4%** | **96.8%** | **91.5%** | **92.1%** | **.719** | .409 |
| (a) | 500 | .160 | .308 | 80.8% | 87.5% | 77.5% | 78.2% | .854 | .499 |
| (b) | 500 | **.179** | **.351** | **87.7%** | **93.0%** | **85.5%** | **85.5%** | **.796** | **.461** |

Table 3: **Quantitative comparison of the limited dataset size.** The above results are from Subject 1. Two scenarios are compared: (a) single subject scenario, where the MindFormer is trained exclusively on subject 1's data, and (b) multi subject scenario, where the MindFormer is trained on data from Subjects 1, 2, 5, and 7. **Bold**: best.

## 4.3 FMRI-TO-TEXT EXPERIMENTS

In order to confirm that the significant improvement of our model is originated from semantically align feature space in MindFormer rather than Stable Diffusion, we additionally perform fMRI-to-text generation experiments by inputting the MindFormer feature as the input of Large Language Model (LLM). This experimental setup is unique as we do not rely on the Stable Diffusion image generator.

**Implementation Details.** Figure 7 shows the framework of the fMRI-to-Text generation. We employ a simple two-layer MLP to align the image feature embeddings from the pretrained Mindformer to the word embedding space of a LLM. We choose OPT-1.3B model as our LLM. With the Mindformer and OPT-1.3B remain frozen, the two-layer MLP is trained with the language modeling loss between the generated captions and the ground truth COCO captions of subjects 1,2,5 and 7. The entire fMRI-to-Image caption model is trained on NVIDIA A100 with 40GB of memory for 5 epochs with a learning rate of 1e-5 and a batch size of 1.

**Results.** To quantitatively compare with other methods, we utilize six text quality evaluation metrics. For assessing low-level properties, we use Meteor (Banerjee & Lavie (2005)), Rouge (Lin (2004)), and CIDEr (Vedantam et al. (2015)). For evaluating higher-level properties, the metrics of SPICE (Anderson et al. (2016)), CLIP (Radford et al. (2021)), and Sentence (Reimers (2019)) are employed. We compared our model with SDReconT (Takagi & Nishimoto (2023b)), UniBrain (Mai & Zhang (2023)), BrainCap (Ferrante et al. (2023)), and MindSemantix (Ren et al. (2024)). We have referenced the result values from the MindSemantix. Table 4 demonstrates superior performance in the metrics of Meteor, CIDEr, SPICE, and Sentence, compared to other models. Also, Figure 8 shows the COCO captions (ground-truth) and generated texts from fMRI signals, and our model successfully generated the caption, corresponding to the stimulus image. The comprehensive results indicate that the output embedding of our MindFormer contains the relevant information needed to generate texts effectively.

## 4.4 DISCUSSION

The results from our experiments show the efficacy and robustness of the MindFormer model in cross-subject brain decoding. One of the key findings is the significant improvement in image reconstruction accuracy when incorporating subject tokens and utilizing a multi-subject training approach. Notably, even with a limited dataset, the MindFormer demonstrates its capability to reconstruct high-quality images, outperforming models trained on larger datasets. This is particularly important given the challenges associated with obtaining large fMRI datasets from new subjects. The ability to achieve good performance with smaller datasets not only validates the efficiency of the MindFormer but also points to its practical applicability in real-world scenarios where data availability may be constrained. Moreover, the comparison between single-subject and multi-subject training further validates the advantage of leveraging data from multiple subjects. The results consistently indicate that the MindFormer approach, which integrates data from subjects 2, 5, and 7 along with subject 1, yields better performance metrics. This suggests that the model benefits from the additional information provided by the diverse set of neural patterns, enhancing its generalization capabilities and robustness. Also, with the output embedding of our MindFormer, LLM can generate the caption

Figure 8: Results of fMRI-to-text model using MindFormer embedding.

| Method | Low-level | | | High-level | | |
|---|---|---|---|---|---|---|
| | Meteor ↑ | Rouge ↑ | CIDEr ↑ | SPICE ↑ | CLIP ↑ | Sentence ↑ |
| SDReconT | 0.100 | 0.251 | 0.138 | 0.050 | 0.624 | 0.280 |
| UniBrain | 0.169 | 0.222 | — | — | — | — |
| BrainCap | 0.167 | 0.407 | 0.413 | 0.091 | 0.705 | 0.447 |
| MindSemantix | 0.190 | **0.415** | 0.476 | 0.125 | **0.755** | 0.454 |
| Ours | **0.291** | 0.358 | **0.634** | **0.177** | 0.744 | **0.485** |

Table 4: **Quantitative results of fMRI-to-text on Subject 1**. **Bold**: best, underline: second best.

corresponding to the stimulus image. It implicitly demonstrates that our Mindformer can be the framework for the fMRI-to-Language generation model.

Although the MindBridge model (Wang et al., 2024) is designed to exploit multi-subject fMRI data for training, it relies on an aggregation function to reduce dimensionality, leading to information loss and lower performance. On the other hand, the learnable subject tokens play a pivotal role in our MindFormer framework. By allowing the model to differentiate between neural signals from different subjects, the subject tokens ensure that the individual-specific nuances in brain activity are preserved and accurately decoded. Table 2 demonstrates the performance improvements attributed to the use of subject tokens, highlighting their importance in achieving high-fidelity decoding outcomes. This approach also streamlines the process by enabling the use of a single unified model for multiple subjects, offering significant advantages in terms of efficiency and scalability.

## 5 CONCLUSION

In this paper, we proposed "MindFormer", a powerful transformer architecture for semantically aligned multi-subject fMRI embedding for braining decoding. Thanks to the effective multi-subject fMRI signal embedding using the subject token and IP-Adapter, the model significantly outperformed the existing multi-subject brain decoding framework. Overall, MindFormer provides a new framework for understanding multi-subject brain decoding and common neural patterns. The model's ability to leverage shared information across subjects while maintaining individual-specific accuracy marks a significant advancement in the field of brain decoding.

**Limitation.** Current implementation of MindFormer primarily focuses on visual stimuli. Extending this approach to decode more complex cognitive and sensory experiences will require substantial advancements in both model architecture and training methodologies. Another limitation is the computational complexity associated with training much more subjects. Although our approach reduces the parameter count compared to existing models, training these models with over about 10 subjects still require significant computational resources. Future work should aim to optimize the model further to make it more accessible and feasible for broader applications.

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

# A APPENDIX

## A.1 COMPARISON OF MODEL PARAMETER NUMBER

| Method | | The number of model parameter |
|---|---|---|
| Brain-Diffuser | Low Level | 1,433,616,800 |
| | High Level | 12,076,800 |
| MindEye | Low Level | 205,505,988 |
| | High Level | 1,003,635,072 |
| MindBridge | single (for 1 subject) | 561,283,712 |
| | multi (for 4 subjects) | 693,579,264 |
| MindFormer | single (for 1 subject) | 304,782,336 |
| | multi (for 4 subjects) | 765,607,680 |

Table 5: MindBridge and MindFormer have fewer parameters compared to other models. In particular, the single-MindFormer for one subject has the fewest number of parameters among them.

## A.2 ADDITIONAL RESULTS OF MULTI MINDFORMER MODEL

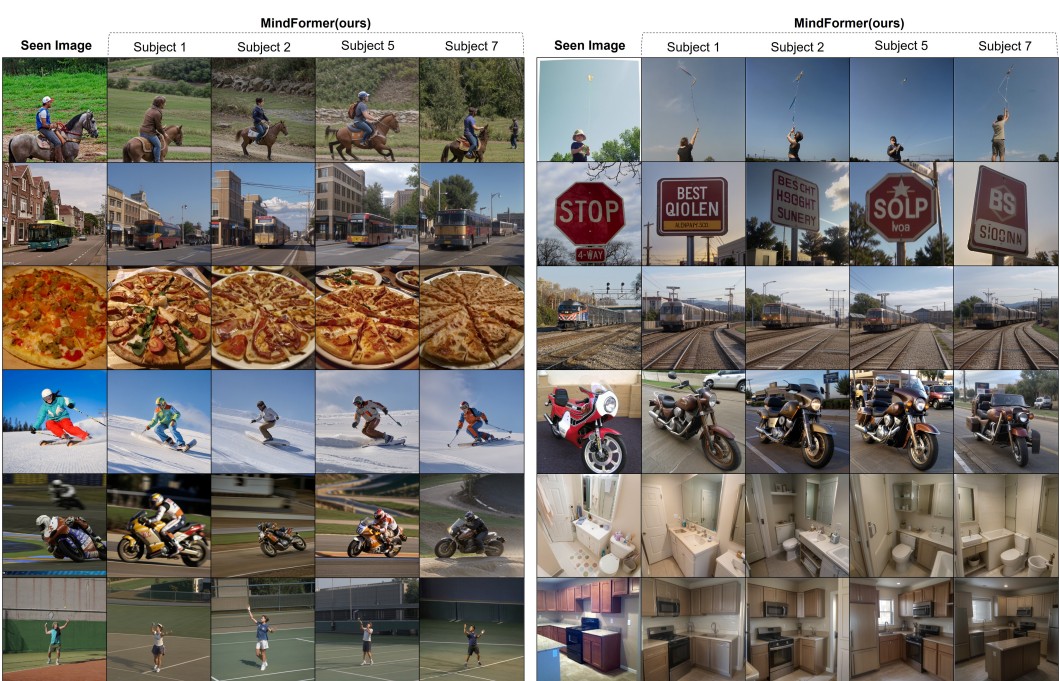

Figure 9: Additional reconstructed image from human brain activity using MindFormer.

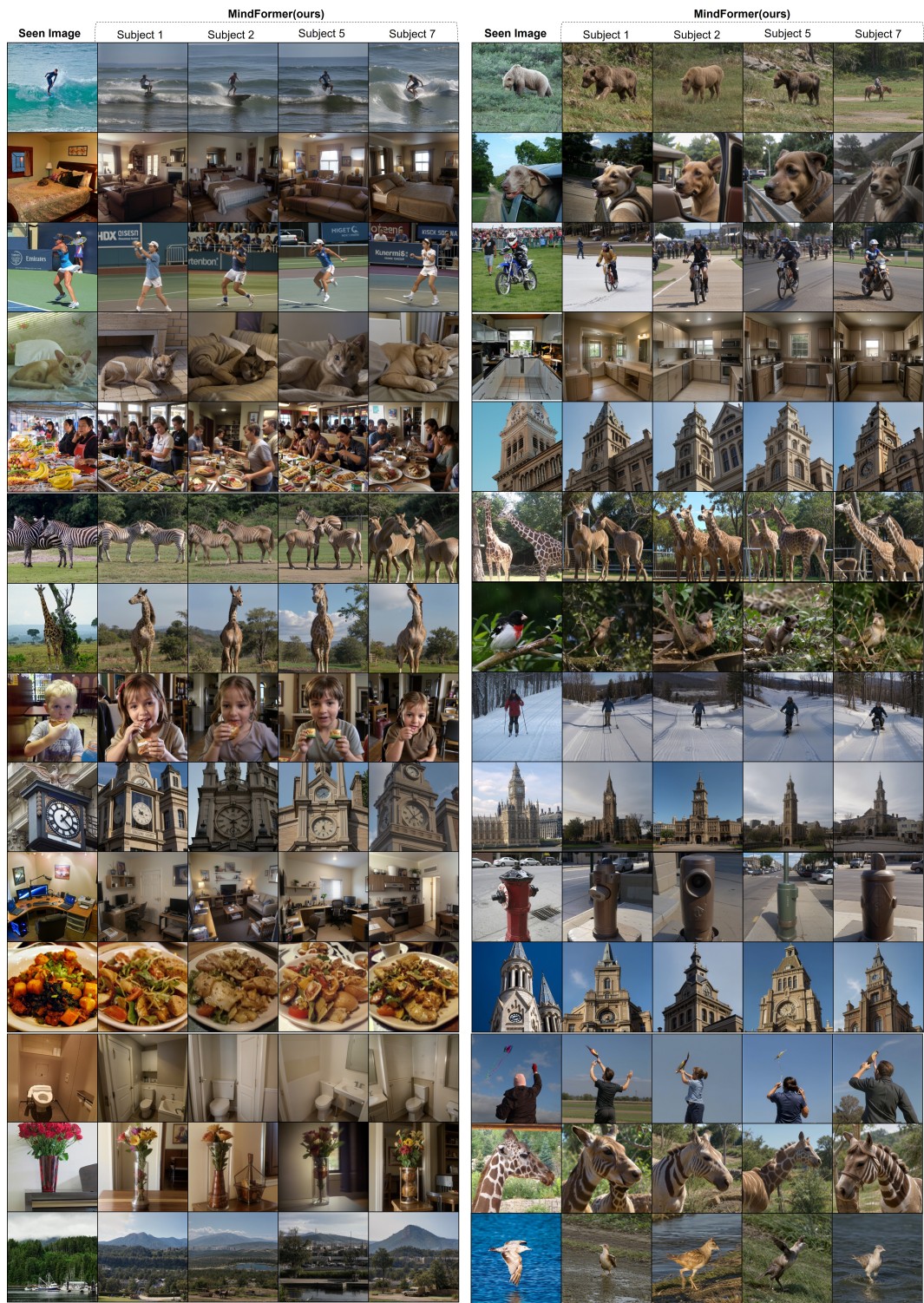

Figure 10: Additional reconstructed image from human brain activity using MindFormer.

