# OpenReview forum: "MindFormer: Semantic Alignment of Multi-Subject fMRI  for Brain Decoding"
_ICLR.cc/2025/Conference — ICLR 2025 Conference Withdrawn Submission_

### Official Review · Reviewer_btih · 2024-10-28

**Soundness:** 2
**Presentation:** 2
**Contribution:** 1
**Rating:** 5
**Confidence:** 4

**Summary:**

This paper introduces MindFormer, a multi-subject fMRI model for brain decoding. The authors achieve this target by aligning fMRI with IP-Adapter's image embedding using subject tokens and subject-specific linear projectors. The evaluations are carried out on the NSD dataset and fMRI-to-image and fMRI-to-text reconstruction are performed. In addition, an ablation study and visualization are also performed.

**Strengths:**

+ Manuscript is clear and easy to understand.
+ The paper gives detailed implementation details.
+ The results presented in this paper are good, especially when the size of the dataset is limited.

**Weaknesses:**

#### 1. The paper fails to provide new insight into this field:
+ At the methodological level,  all key designs in this paper have been presented in existing papers. Specifically, CLIP-MUSED [1] proposed subject tokens, and MindEye2 [2] proposed subject-specific linear projectors.
+ At the level of interpretability, the authors do not provide further valid the mechanisms by which these methods are effective.

#### 2. Lack of validation of model lightweight
In Line 096, the authors mentioned their model was smaller and more efficient. However, there are no experiments in this paper to show these results. Since researchers in this field have paid little attention to model efficiency issues, further research on model lightweight would add to the paper's contribution. The number of model parameters, the memory overhead during inference, and the time needed for reconstruction are metrics that should be considered.

#### 3. Lack of rigor in writing
The authors should review and clarify some error points, including but not limited to:
+ Equation 3 is missing a negative sign.
+ In Figure 3, the starting point of connecting lines between training and inference is not clear.

[1] Qiongyi Zhou et al. CLIP-Guided Multi-Subject Visual Neural Information Semantic Decoding. ICLR 2024

[2] Paul S. Scotti et al. MindEye2: Shared-Subject Models Enable fMRI-To-Image With 1 Hour of Data. ICML 2024

**Questions:**

+ According to the description of methods, this paper appears that only high-level reconstruction was performed, does this mean that low-level reconstruction in the MindEye series of work is not necessary?
+ As far as I know, the pre-trained Stable Diffusion doesn't seem to be able to receive 16×768 image embedding as a condition, how is image reconstruction implemented? The authors should provide more details on how they adapted or modified the Stable Diffusion.
+ This paper uses the **nsdgeneral** ROI, which primarily covers the brain's visual cortex. does it make sense to use this ROI for fMRI-to-text reconstruction?

---

### Official Review · Reviewer_zSme · 2024-11-03

**Soundness:** 2
**Presentation:** 3
**Contribution:** 1
**Rating:** 1
**Confidence:** 4

**Summary:**

The paper introduces a novel method for fMRI-to-image reconstruction, leveraging IP-adapters to extract semantic features from fMRI signals. The authors evaluate their approach in a multi-subject setting and perform various experiments with limited data. Additionally, they test their proposed pipeline in an fMRI-to-text scenario to demonstrate the presence of a semantically aligned feature space within MindFormer.

**Strengths:**

Among the strengths of the paper, the introduction of learnable subject tokens stands out as a significant contribution to the proposed method. Additionally, the dimensionality of the embeddings output by the IP-adapter appears to help reduce model size, which could be particularly advantageous when scaling up to accommodate a larger number of subjects, though further investigation on this is warranted. Finally, achieving strong performance in fMRI-to-text generation with only a limited number of fine-tuning epochs is a notable advantage of the proposed method.

**Weaknesses:**

1) The paper claims that the use of the IP-adapter significantly reduces the overall model size. However, there is no comparison provided between the final size of the proposed method and the final sizes of the models it is benchmarked against. Adding this comparison would clarify the actual efficiency gains achieved by the IP-adapter.

2) The main competitor to this paper is the MindEye2 (Scotti et al., 2024) approach, and a comparison with this method is essential to validate the effectiveness of the proposed method. In terms of experiments conducted in the limited data setting, there appears to be a lack of novelty, as MindEye2 has demonstrated significantly better results with fewer data. Including a comparative analysis here would strengthen the argument for the proposed method’s effectiveness. Finally, regarding the linear projection used in the proposed approach, there is limited novelty, as MindEye2 also employs a subject-specific ridge regression layer to project fMRI signals into a common representational space. This similarity suggests that the contribution of the linear projection may not be unique to this work.

**Questions:**

Addressing the issues raised in the 'Weaknesses' section would indeed be sufficient to improve the paper. By clarifying these points, the authors can address potential gaps and enhance the overall impact and rigor of their findings.

---

### Official Review · Reviewer_Y8cd · 2024-11-04

**Soundness:** 3
**Presentation:** 4
**Contribution:** 3
**Rating:** 6
**Confidence:** 4

**Summary:**

Mindformer maps a representation of fMRI data to images in a multi-component architecture. The work has the main goal of recalling visual-images/text from synchronized fMRI data. They suggest a way to condition fMRI data on subjects with a trained subject specific token. Conditioned on subjects, a transformer model encodes input fMRI features to output feature vectors. They adopt the IP adapter, that is trained to match image feature embeddings with transformer output feature vectors. With IP adapter, they generate a significantly smaller feature vector compared to CLIP embedding.
Latent diffusion model embeds visual-images/text into a latent space conditioned on the output feature vectors of the transformer and the IP adapter image feature embeddings.
They evaluate the suggested model on a common benchmark, Natural Scenes Dataset. Multi-subject decoding results and low-data regime  cases are evaluated on low-level (PixCorr, SSIM, AlexNet) and high level (Inception, CLIP, EffNet-B, SwAV) metrics.
Another evaluation is on a large language model that replaces the latent diffusion model, where they measure the recall performance of the generated image captions.
Mindformer has higher performance in most of the selected metrics for image recall experiments and image-caption recall experiments.

**Strengths:**

1- The ablation of learnable subject specific token and improved performance with multiple subjects is convincing.

2- The study has higher performance in most of the visual and textual similarity metrics.

3- The framework implementation of Mindformer for image and text recall is sound and convincing.

**Weaknesses:**

1- fMRI-space visualization is missing. Deep models are difficult to interpret, mainly due to high nonlinearity. To gain insights about "understanding multi-subject brain decoding and common neural patterns", an experiment on visualizing model saliency in fMRI-space is missing, especially to investigate common neural patterns. For instance, the study can investigate the role of different areas in the visual cortex on the model performance.

2- Latent space analysis can further support the claims of the study. The experiments focus too much on the performance improvement aspect with multiple subjects. Visualizing the latent space (visual image and fMRI data pair) distribution that highlights the change of common and subject-specific clusters of the same semantic category, if any, can improve the study.

3- A minor weakness is the limited novelty in the as-is adopted components from related studies.

**Questions:**

1- Would it be possible to visualize the fMRI-space saliency of MindFormer?

2- Can we also quantify on the latent space distribution and support the claim that multi-subject data is beneficial for the model, as it aligns fMRI data on the common visual concepts?

3- A discussion on the difficulties of the blackbox interpretation of the proposed model can improve the completeness of the study, towards gaining an understanding based on the outcomes of the model.

---

### Official Review · Reviewer_V69u · 2024-11-04

**Soundness:** 2
**Presentation:** 2
**Contribution:** 2
**Rating:** 3
**Confidence:** 4

**Summary:**

This paper proposed a transformer architecture for semantically aligned multi-subject fMRI embedding for braining decoding, called MindFormer. Experimental results demonstrate that this method maintains strong performance of fMRI reconstruction,even with limited data, by introducing IP-Adapter and subject token.

**Strengths:**

1. The basic methodology of the paper is clearly stated and reproducible.
2. This proposed method seems to be effective, achieving better performance.
3. Extensive experiments are conducted to analyze the effectiveness of the proposed method.

**Weaknesses:**

1. Some accounts are too subjective and lack appropriate supporting information. For example, authors state in lines 175-178 that 'However, this variability in Versatile Diffusion can introduce challenges in brain decoding, as the reconstructed images may display inconsistencies and artifacts. These discrepancies can complicate the accurate interpretation of neural activity, potentially reducing the fidelity of the decoding outcomes.' The assertion lacks any supporting evidence in the relevant literature, and the flaws in the generative model appear to be inconsistent with the motivation for this paper.
2. Insufficient discussion of related work. There are already many published papers discussing multi-subject models [1, 2]. I think that even if quantitative comparisons are not made for reasons of timeliness, they need to be mentioned and discussed in related work.
3. The novelty of this paper is limited in some ways. As rendered in this article, MindFormer's approach is more like incremental work. Small modular innovations are present, e.g. modifying the design of the alignment embeddings, but do not contribute much to the overall decoding task (since the baseline paper has pioneered multi-subject decoding). Personally, I think the innovation is more applicable to the traditional computer vision field than to the brain decoding field.


**Reference**
1. Scotti, Paul S., et al. "MindEye2: Shared-Subject Models Enable fMRI-To-Image With 1 Hour of Data." ICML 2024.
2. Shen, Guobin, et al. "Neuro-Vision to Language: Image Reconstruction and Interaction via Non-invasive Brain Recordings." NeurIPS 2024

**Questions:**

1. The author mentions that MindFormer is efficient as a major contribution, but the main text does not show any quantitative results. And '257×768 and 77×768' makes reader confused when first mentioned. From past research as well as scaling laws, shouldn't the embeddings of larger models be better?
2. It is doubtful that the IP-Adapter is better than previous models like ViT. Are there any relevant ablation experiments that indicate this?
3. The results in Table 3 show that (a)'s low-level performance is better than (b)'s when the dataset is larger, is there any explanation for this?

---

### Note · Authors · 2024-11-13

I have read and agree with the venue's withdrawal policy on behalf of myself and my co-authors.